# No Effect of Cathodal Transcranial Direct Current Stimulation on Fear Memory in Healthy Human Subjects

**DOI:** 10.3390/brainsci6040055

**Published:** 2016-11-04

**Authors:** Aditya Mungee, Max Burger, Malek Bajbouj

**Affiliations:** 1Department of Psychiatry, Campus Benjamin Franklin, Charité, Berlin 14050, Germany; max-benedict.burger@charite.de (M.B.); malek.bajbouj@charite.de (M.B.); 2Cluster of Excellence “Languages of Emotion”, Freie Universität, Berlin 14195, Germany; 3Dahlem Institute for Neuroimaging of Emotion, Freie Universität, Berlin 14195, Germany

**Keywords:** direct current stimulation, fear conditioning, memory, prefrontal cortex

## Abstract

Background: Studies have demonstrated that fear memories can be modified using non-invasive methods. Recently, we demonstrated that anodal transcranial direct current stimulation (tDCS) of the right dorsolateral prefrontal cortex is capable of enhancing fear memories. Here, we examined the effects of cathodal tDCS of the right dorsolateral prefrontal cortex during fear reconsolidation in humans. Methods: Seventeen young, healthy subjects were randomly assigned to two groups, which underwent fear conditioning with mild electric stimuli paired with a visual stimulus. Twenty-four hours later, both groups were shown a reminder of the conditioned fearful stimulus. Shortly thereafter, they received either tDCS (right prefrontal—cathodal, left supraorbital—anodal) for 20 min at 1 mA, or sham stimulation. A day later, fear responses of both groups were compared. Results: On Day 3, during fear response assessment, there were no significant differences between the tDCS and sham group (*p >* 0.05). Conclusion: We conclude that cathodal tDCS of the right dorsolateral prefrontal cortex (right prefrontal—cathodal, left supraorbital—anodal) did not influence fear memories.

## 1. Introduction

Different interventions have been studied as possible pathways to modify fear memories in animals as well as humans, since this could lead to new treatment methods for anxiety disorders and post-traumatic stress disorder (PTSD). The reconsolidation window has been a popular target for manipulating fear memories, since consolidated memories return to a labile state after reactivation, and reconsolidation requires de novo protein synthesis in the amygdala [1]. In animals, protein synthesis inhibitors like anisomycin have been used to inhibit fear memory reconsolidation [2]. Glucocorticoids have also been shown to impair fear memory reconsolidation in mice; this effect depended on the muscarinic cholinergic receptors, and was blocked by atropine [3]. A recent study showed that ketamine—a *N*-methyl-d-aspartate receptor antagonist—impaired reconsolidation of contextual fear memory in rats [4].

There have also been pharmacological studies to diminish fear in humans. Propranolol has been shown to interfere with fear and trauma memory reconsolidation [5]; however, these effects have not been consistently replicated [6]. Cortisol has been shown to influence fear reconsolidation in men, but this effect could not be replicated in women [7]. Aside from such pharmacological interventions, various non-invasive methods have also been studied to interfere with fear memories. Schiller et al. used the reconsolidation window to behaviourally rewrite fear memories with non-fearful information [8]. Burger et al. showed that tVNS (transcutaneous vagus nerve stimulation) accelerates explicit fear extinction; however, it did not lead to better retention of extinction memory 24 h later [9]. A common problem with most of these techniques is the lack of replicability and consistent long term effects. Hence, it is important to systematically investigate different fear modulating interventions. 

Non-invasive brain stimulation—more specifically, transcranial direct current stimulation (tDCS)—has also been shown to influence fear memories [10]. tDCS has been shown to alter cortical excitability; anodal tDCS results in neuronal depolarisation, leading to an excitatory effect, whereas cathodal tDCS results in hyperpolarisation, and thus has an inhibitory effect [11]. Based on these antagonistic effects, many studies have demonstrated that tDCS affects cognition, mood, and working memory [12]. 

To date, three studies targeting different processes have demonstrated that tDCS can affect fear memories in humans. Van’t Wout et al. [13] stimulated the left dorsolateral prefrontal cortex (DLPFC) with anodal tDCS during extinction learning, and reported enhanced subsequent extinction of conditioned fear. Asthana et al. [14] reported that cathodal tDCS of the left DLPFC resulted in inhibition of fear memory consolidation, whereas anodal tDCS did not lead to the enhancement of fear memory. Previously, we have shown that tDCS (right prefrontal—anodal, left supraorbital—cathodal) during the reconsolidation window resulted in enhancement of fear memories [10]. The fear circuit underlying memory reconsolidation is unclear, and finding a window for tDCS to manipulate this circuit is challenging. It has been proposed that noradrenergic and glutamergic inputs to the basolateral nucleus of the amygdala (BLA) might restabilize glutamate-dependent plasticity during reconsolidation [15]. Furthermore, the authors propose that BLA output neurons to the central nucleus of the amygdala control aversive responses during emotional memory expression. tDCS can only induce changes in cortical excitability, and effects on deeper areas like the amygdala are indirect [16]. Hence, we chose to target the prefrontal cortex–amygdala circuit to disrupt fear memory reconsolidation. It is known that the DLPFC is involved in cognitive regulation of conditioned fear, and projects to the ventromedial prefrontal cortex (vmPFC), which in turn has inhibitory connections to the amygdala [17]. Since our previous study influencing the same neural circuit resulted in the enhancement of fear memories [10]—in line with the AeCi hypothesis (anodal: excitatory, cathodal: inhibitory)—we reversed this protocol and performed right prefrontal—cathodal, left supraorbital—anodal tDCS during the reconsolidation window. We chose the right dorsolateral prefrontal cortex as our target region for inhibitory cathodal stimulation because it is known that the right prefrontal cortex is activated during the processing of negative emotions [18]. The reference electrode for anodal stimulation was positioned on the left supraorbital area, close to the ventromedial prefrontal cortex, which possibly has an inverse relationship with the amygdala [19]. Enhancing the inhibitory connections from the vmPFC to the amygdala through anodal tDCS could inhibit fear memory. Hence, we propose that inhibitory stimulation of the DLPFC, together with excitatory stimulation of the vmPFC, would result in the inhibition of fear memories. 

## 2. Methods and Materials

We followed an identical protocol as in our previous study [10], but for one important change; the electrode positions of the anode and cathode were reversed during tDCS stimulation in order to achieve the opposite effect.

### 2.1. Subjects

Twenty-five healthy subjects aged between 18 and 40 years were recruited by poster advertisements for the study. Subjects with metal implants inside the skull or eye, severe scalp skin lesions, cranial bone fractures, known history of epilepsy or previous seizures, pregnant or breast-feeding women, and subjects with a known psychiatric disorder or on CNS-acting medications were excluded from the study. The Charité institutional ethics committee approved the protocol, which was conducted in accordance with the Declaration of Helsinki. All subjects were given a complete oral and written description of the study, and informed consent was obtained from each subject prior to participation. The subjects were compensated for their participation in the study. 

### 2.2. Procedure 

The experiment was conducted over 3 days, with a wash-out period of 24 h between the sessions. A quiet room was used to conduct the sessions in order to minimize the influence of external stimuli on skin conductance. The protocol for all three sessions is summarized in Table 1. The experiment was performed using Presentation^®^ software (Version 7.0, [20]).

Two randomly assigned groups (tDCS and sham) underwent a Pavlovian fear conditioning paradigm with partial reinforcement. The conditioned stimuli (CS) were blue and yellow squares, and the unconditioned stimulus (US) was a low intensity electric shock to the wrist. One stimulus was paired with the US on 38% of the trials (CS+), and the other was never paired with a shock (CS−). We used a partial reinforcement schedule to avoid rapid extinction [7]. A Grass Medical Instruments stimulator was used to deliver 50 pulses/second for a duration of 200 ms. The intensity of the electric shock was adjusted to every individual subject, the threshold stimulus being uncomfortable but not painful. We used a starting stimulus of 10 V, and went up to a maximum intensity of 60 V. The participants were shown 10 randomized presentations of the CS+ and CS− each, and additionally 6 CS+ presentations, which were associated with a shock (US). The order of appearance of the colour paired with the shock was also randomized in order to avoid bias. We presented the stimuli for 4 s, with a 10–12 s gap between stimuli. The duration between stimuli was randomized in order to avoid false responses because of habituation. Skin conductance responses were measured using the Schuhfried Biofeedback X-pert 2000 device (Schuhfried, Moedling, Austria). The electrode was connected to the ring finger of the left hand.

#### 2.2.1. Day 2: tDCS

On the second day all subjects were shown a reminder using a single presentation of the coloured square (CS+) which was paired with a shock on Day 1. However, no shock was administered on Day 2. Immediately after this, the subjects in the tDCS group were stimulated with tDCS (1 mA) for a total duration of 20 min using two saline-soaked surface sponge electrodes (15 cm^2^) with a current density of 0.67 A/m^2^. The subjects in the sham group received only a brief current for the first 30 s in order to mimic the itching associated with real stimulation. The cathodal electrode was placed on the region of interest, the right DLPFC with electrodes (5 × 3 cm) placed at the right frontolateral location (F4 of the international 10:20 electroencephalogram system) [21], and the anode on the contralateral supraorbital area. We used a constant current battery driven stimulator (CX6650, Rolf Schneider Electronics, Gleichen, Germany). The current was ramped up to 1 mA over a period of 30 s to minimize side effects. 

#### 2.2.2. Day 3: Fear Response Assessment

To test the effect of tDCS, we presented both groups with the conditioned stimulus without the unconditioned stimulus to assess their fear responses on Day 3. We presented the subjects with 10 CS+ and 11 CS− presentations, and the order of appearance was randomized to prevent bias. Since the subjects were shown a reminder of CS+ on the second day, we used one extra presentation of CS− on Day 3 in order to keep the total number of trials on all three days equal.

### 2.3. Data Analysis

The response to an average of all trials during fear acquisition on Day 1 (except the first trial) was used as a criterion to decide if the subjects had successfully acquired fear conditioning. It has been proposed that human participants habituate to the CS+ during conditioning, and that fear responses might be better observed in the earlier CS+ trials [22]. We first assessed fear conditioning in all subjects (*n* = 25) before discarding any data. Subjects (*n* = 8) where CS+ was equal or less than CS− for an average of all the trials during acquisition were excluded from the analysis. Hence, 17 subjects were included in the final sample [tDCS = 7 (*m* = 3, *f* = 4); sham = 10 (*m* = 2, *f* = 8)]. We used Ledalab, a MATLAB (Mathworks Inc., Sherborn, MA, USA) based software, more specifically the CDA (Continuous Decomposition Analysis) method to analyse the skin conductance data. This method extracts the phasic information underlying the skin conductance response, and aims at retrieving the signal characteristics of the underlying sudomotor nerve activity [23]. Since we expected the fear responses to be most pronounced in the early phase on Day 3, we restricted our analysis to the first three presentations of the CS+ and CS−. Since approximately one-third of the CS+ trials were paired with a shock (US) on the first day, we expected the conditioned subjects to show a fear response to at least the first three trials of CS+ on the third day. However, since no shocks are actually administered, a gradual learning effect and thus diminishing of the fear responses is expected after the early phase. We compared the mean differential SCR (skin conductance response) between the tDCS and the sham groups in the 0.5 to 4.5 s time window after stimulus onset (CS+ minus CS−). Square root transformation of the raw data was performed to normalize distributions. Each subject’s normalized score was then divided by the mean square-root-transformed US response of that subject. Statistical analysis was performed using SPSS 20 (SPSS Inc., Chicago, IL, USA)

## 3. Results

All subjects tolerated the tDCS stimulation well, and no adverse effects were reported.

### 3.1. Day 1—Fear Acquisition

Fear responses were analysed for all subjects in the late phase on Day 1 (last three CS+ and last three CS− trials) using a repeated measures ANOVA with CS as the within-subjects factor and group (tDCS/sham) as the between-subjects factor. We found significant main effects of CS trial [F(1,15) = 15.22, *p* = 0.001, η*p*^2^ = 0.50], but no significant effects for group [F(1,15) = 1.09, *p* > 0.05, η*p*^2^ = 0.07], or the interaction between group and CS [F(1,15) = 4.19, *p* > 0.05, η*p*^2^ = 0.22]. The significant difference between CS+ and CS− indicates that participants successfully acquired fear conditioning on Day 1, and there was no significant difference between the sham and the real group. 

### 3.2. Day 3—Fear Memory Test

We first analysed the first three CS+ and the first three CS− trials on Day 3, since we expected gradual habituation after the initial phase. There were no significant effects for CS [F(1,15) = 2.05, *p* > 0.05, η*p*^2^ = 0.12] or group (tDCS/sham) [F(1,15) = 3.38, *p* > 0.05, η*p*^2^ = 0.18]; the interaction between CS and group was also not significant [F(1,15) = 0.55, *p* > 0.05, η*p*^2^ = 0.04]. Next, we conducted a repeated measures ANOVA for only the first two trials of CS+ and CS− on Day 3. We found significant effects for CS [F(1,15) = 5.28, *p* < 0.05, η*p*^2^ = 0.26], but no significant effects for group [F(1,15) = 3.60, *p* > 0.05, η*p*^2^ = 0.19] or the interaction between CS and group [F(1,15) = 2.15, *p* > 0.05, η*p*^2^ = 0.13]. These results indicate that the participants showed defensive responses up to the first two trials of CS+ and CS− each. Subsequently, rapid habituation took place so that there were no significant defensive responses after three trials each of CS+ and CS−. There was no significant effect of tDCS on this habituation; both the tDCS and the sham groups showed habituation after 3 trials each of CS+ and CS−.

## 4. Discussion

In this study, we investigated the effects of tDCS (right prefrontal—cathodal, left supraorbital—anodal) on fear memories. Our results show no significant differences in the fear response between the tDCS and sham group. Recently, we reported that tDCS (right prefrontal—anodal, left supraorbital—cathodal) results in enhancement of fear memories [10]. According to the assumption that anodal tDCS enhances cortical excitability, while cathodal tDCS diminishes it [11], we hypothesized that reversing this stimulation protocol would result in an inhibition of fear memories. However, our results do not confirm this hypothesis. 

There are two possible explanations why tDCS (right prefrontal—cathodal, left supraorbital—anodal) did not inhibit fear memories. Firstly, because the dual-polarity effect (more specifically, the cathodal inhibitory effect) is more difficult to replicate while investigating memory in comparison to motor cortex [24]. The hypothesis that anodal tDCS promotes cortical excitability while cathodal tDCS inhibits it has been established mainly through motor cortex studies [25]. It has been proposed that the lack of a cathodal inhibition effect in cognitive studies could be state-dependent due to the brain areas involved already being highly activated during a task [24]. This explanation is plausible in the case of our study; since the participants saw a reminder of the conditioned stimulus (connected to the electrical shock) just before tDCS, this probably resulted in the fear circuit being activated during the stimulation, hence making it difficult for the inhibitory cathodal stimulation to diminish these reactivated fear memories, as opposed to enhancement through anodal tDCS, where reactivation might play a facilitatory role [10]. Previous cognitive studies have also highlighted the difficulty in replicating polarity-dependent effects. Marshall et al. [26] reported that both cathodal and anodal tDCS intermittent stimulation of the lateral prefrontal cortex during a working memory task slowed reaction time. Kincses et al. [27] reported that anodal tDCS improved implicit learning; however, cathodal tDCS did not significantly impair implicit learning. Our results are in line with these findings, since cathodal tDCS of the prefrontal cortex had no effect on fear memories, while anodal tDCS has been shown to enhance fear [10]. 

Secondly, it is possible that the complex fear neural circuitry involved in reconsolidation is difficult to manipulate by stimulating superficial cortical areas like the DLPFC and vmPFC with a single session of tDCS, as opposed to deeper subcortical regions like the amygdala; however, these deep regions are technically challenging to stimulate using tDCS. At best, tDCS is capable of indirectly modifying cortico-subcortical connections [16]—a single session of tDCS might not be enough to manipulate physiological defensive responses reliably and consistently. To our knowledge, there are no published studies on the effect of repetitive sessions of tDCS on fear memory reconsolidation. Recently, LeDoux et al. [28] proposed a two systems framework with the first circuit involving cortical areas which are responsible for generating feelings of fear and anxiety, and a second circuit involving subcortical areas like the amygdala which are responsible for behavioural and physiological responses to fear. Further, it is suggested that the amygdala itself is not a fear center generating the experience of fear, but rather responsible for detecting and responding to threats. According to this model, stimulating cortical regions like the DLPFC and the vmPFC with tDCS might have more influence on the subjective feeling of fear rather than the physiological response to fear, since the second circuit is not directly influenced by tDCS. On similar lines, Schiller et al. [29] reported that while standard extinction involves areas of the prefrontal cortex like the vmPFC, extinction during the reconsolidation window appears to bypass the prefrontal cortex. Taking these findings together, we recommend that future studies with manipulations during the reconsolidation window should try to develop new techniques to target the amygdala, rather than the prefrontal cortex.

It is important to carefully consider the differences between our protocol and earlier studies which have manipulated fear using tDCS. Firstly, the studies differed on the target and reference electrode positioning for tDCS. Asthana et al. [14] stimulated the F3 area according to the 10–20 EEG system corresponding to the left DLPFC with cathodal and anodal tDCS in two different groups; the reference electrode was located over the left mastoid. Van’t Wout et al. [13] stimulated the AF3 region with anodal tDCS, and the reference electrode was on the contralateral mastoid. The authors argue that this protocol allows the best stimulation of the vmPFC. In our earlier study [10], we stimulated the F4 region corresponding to the right DLPFC with anodal tDCS and the left supraorbital area with cathodal tDCS. Secondly, let us consider differences in current density and the intensity and duration of tDCS. Van’t Wout et al. stimulated with 2 mA, because the target and reference electrodes were situated further away from each other and the authors hoped to stimulate deeper areas with a higher intensity. Asthana et al. and Mungee et al. applied 1 mA intensity for the stimulation. While our duration of stimulation was 20 min, Asthana et al. applied tDCS for 12 min, and Van’t Wout et al. for 10 min. Current density differed between the studies, with Van’t Wout et al. reporting the highest value at 1.33 A/m^2^, Mungee et al. at 0.67 A/m^2^, and Asthana et al. at 0.29 A/m^2^. Fourthly, the nature of the US also differed between the studies. While Asthana et al. used an auditory stimulus (screaming), Van’t Wout et al. and our study used an electric shock to condition the participants. Finally, the studies targeted different pathways to influence fear. Both our current and previous study targeted reconsolidation 24 h after fear acquisition; hence, we performed tDCS during the reconsolidation window after reactivating the memory with a reminder. On the other hand, Asthana et al. performed tDCS during the consolidation phase a few minutes (10–20 min) after fear acquisition on the first day itself. Van’t Wout et al. targeted extinction of conditioned fear, and hence performed tDCS during extinction learning. 

These contrasting stimulation protocols highlight the challenges that we face while attempting to influence different emotional processes with a complex neural circuit through tDCS. Replication studies are needed for targeting each of these pathways to manipulate fear to check for consistent effects, before a conclusive statement can be made on a standard protocol that offers the best pathway to modify processes like fear extinction, consolidation, or reconsolidation. 

Our study is limited by the small sample size (*n* = 17) and gender predominance towards females. This makes a direct comparison with our first study difficult, since we had a larger sample size (*n* = 50) and better gender balance [10]. Because of the relatively large electrode sizes used for tDCS, we cannot rule out the possibility of having stimulated other cortical areas involved in the neural circuit modulating fear. Since we used a bipolar stimulation design, we cannot discern between the effects of the left and right prefrontal electrodes. We used skin conductance response to measure fear, which is susceptible to noise due to spontaneous fluctuations in SCR that constitute within-subject variance [30]. We tried to minimize noise by using the CDA (Continuous Decomposition Analysis) method to analyse SCR; nevertheless, we cannot rule out the effect of residual noise. We did not measure additional physiological responses like heart rate or respiratory rate. On Day 2, we did not measure skin conductance after showing the reminder to measure fear response before tDCS. 

## 5. Conclusions

In summary, we found no effect of tDCS (right prefrontal—cathodal, left supraorbital—anodal) on fear memories, in contrast to our earlier study [10] where we found that tDCS (right prefrontal—anodal, left supraorbital—cathodal) resulted in the enhancement of fear memories. Using alternative protocols targeting other pathways to manipulate fear, Asthana et al. [14] and van’t Wout et al. [13] reported an inhibition of fear memories. To our knowledge, no negative studies with regard to tDCS and fear memories have been published yet. Our results emphasize the need to conduct more studies with diversity in target electrode positions, reference electrode positions, laterality, and intensity in order to identify appropriate stimulation protocols for influencing fear memories. The lack of such studies in healthy participants makes it difficult to develop standard stimulation protocols for patients with PTSD or anxiety disorders. 

## Figures and Tables

**Table 1 brainsci-06-00055-t001:** Overview and timeline of the experiment. tDCS: transcranial direct current stimulation.

Day 1	Day 2	Day 3
Fear acquisition	Group 1 → tDCS (cathodal) [F4]	Fear response assessment
	Group 2 → tDCS (sham) [F4]	

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
