# Peer review of "No Effect of Cathodal Transcranial Direct Current Stimulation on Fear Memory in Healthy Human Subjects"

_brainsci, 2016, doi:10.3390/brainsci6040055_

Round 1

Reviewer 1 Report

The goal of this manuscript was to examine the effects of cathodal tDCS of the right dorsolateral prefrontal cortex on fear memories in humans. The authors followed an identical protocol as in their previous study, but while reversing the electrode positions of the anode and cathode during tDCS stimulation in order to achieve the effect of inhibition of fear memories. They recruited 25 healthy subjects aged between 18 and 40 years. However, 11 subjects were subsequently excluded from the analysis, because in these subjects CS+ was equal or less than CS- during the last trial of acquisition. Considering the results, the authors found no significant differences in the fear response between the tDCS and sham group. One can understand a negative result, however previous studies reported an inhibition of fear memories using alternative protocols.

I think there are several important strengths to this paper: First, the idea of this work is interesting and pertinent. Second, I believe that the presented results represent a potential useful knowledge to the literature. Third, strengths of this study include an adequate experimental design, and the use of appropriate methods to evaluate the studied responses. Moreover, the manuscript is clearly written.

The study did not show any significant difference in the fear response between the tDCS and sham group, after the tDCS procedure. Few comments:

·       The reminder was shown only briefly using a single presentation of the colored square immediately before the tDCS procedure. We do not have evidence that this brief reminder is enough to generate the fear response in participants.

·       The authors should discuss the fact that we have here only one objective measure, SCR, and that it could be interesting to have different ones such as respiratory rate or heart frequency as other potential measures.

·       Another aspect is that participants received only one tDCS session of 20 min failing to inhibit fear memories. It may be interesting to discuss the potential role of repetitive sessions on inhibition of fear memories, particularly if we consider that contrary to the anodal tDCS, cathodal tDCS did not enhance fear memories. 

Author Response

Thank you for the valuable feedback and comments which we integrated in the revised manuscript. Please find below a point-by-point-reply.

The goal of this manuscript was to examine the effects of cathodal tDCS of the right dorsolateral prefrontal cortex on fear memories in humans. The authors followed an identical protocol as in their previous study, but while reversing the electrode positions of the anode and cathode during tDCS stimulation in order to achieve the effect of inhibition of fear memories. They recruited 25 healthy subjects aged between 18 and 40 years. However, 11 subjects were subsequently excluded from the analysis, because in these subjects CS+ was equal or less than CS- during the last trial of acquisition. Considering the results, the authors found no significant differences in the fear response between the tDCS and sham group. One can understand a negative result, however previous studies reported an inhibition of fear memories using alternative protocols.

I think there are several important strengths to this paper: First, the idea of this work is interesting and pertinent. Second, I believe that the presented results represent a potential useful knowledge to the literature. Third, strengths of this study include an adequate experimental design, and the use of appropriate methods to evaluate the studied responses. Moreover, the manuscript is clearly written.

The study did not show any significant difference in the fear response between the tDCS and sham group, after the tDCS procedure. Few comments:

The reminder was shown only briefly using a single presentation of the colored square immediately before the tDCS procedure. We do not have evidence that this brief reminder is enough to generate the fear response in participants.

We acknowledge the reviewer’s concerns that we do not have evidence in form of skin conductance data from Day 2 after showing the reminder to objectively state that the participants showed a fear response to the stimulus. We have added this as a limitation of our study (lines 260-261).

The authors should discuss the fact that we have here only one objective measure, SCR, and that it could be interesting to have different ones such as respiratory rate or heart frequency as other potential measures.

We agree with the reviewer that our study has only one outcome measure (skin conductance), which is a limitation that we have added to the manuscript (lines 259-260). Retrospectively, it would have been helpful to have other measures like respiratory rate or heart frequency. We acknowledge the reviewer’s suggestion for future studies.

Another aspect is that participants received only one tDCS session of 20 min failing to inhibit fear memories. It may be interesting to discuss the potential role of repetitive sessions on inhibition of fear memories, particularly if we consider that contrary to the anodal tDCS, cathodal tDCS did not enhance fear memories.

We have expanded the discussion section to include the neuroanatomy of the neural circuit underlying reconsolidation and mentioned the technical challenges of stimulating the amygdala through the prefrontal cortex-amygdala connections with just one session of tDCS (lines 204-222). Future studies should consider performing repetitive tDCS to  increase the depth and efficacy of stimulation.

Reviewer 2 Report

Manuscript ID: brainsci-153866

Title: No effect of cathodal transcranial direct current stimulation on fear memory in healthy human subjects.

The authors present a study in which they aim to inhibit defensive responses after fear reconsolidation in a Pavlovian fear conditioning paradigm using cathodal tDCS. This is in essence the opposite from an earlier study published by the authors where they used anodal tDCS to enhance reconsolidation after fear conditioning. Contrary to their previous finding of anodal tDCS, cathodal tDCS after fear reconsolidation did not seem to affect defensive responses 24 hours later. I think the publication of null results like this are important, because this would ultimately result in defining the parameters to allow better targeting of brain circuitry and processes for ultimate improved treatment of specific mental health problems. However, there are a number of issues I think the authors should address before I can recommend publication. In a way my comments are mostly related to 1) thinking about the data, analyses and subsequent interpretation, and 2) comparisons with prior studies – as that would allow a paper with non-significant results to have the largest impact on the field and accomplish the goal above.  

Broad comments:

1.       Although the authors recognize and mention their small sample size; a total of 14 people - 7 randomized to active tDCS and 7 randomized to sham, this does remain a critical issue when dealing with interpreting results. I won’t ask for retrospective power analyses as I tend to find these circular in nature. However, I think it might be worthwhile to add standardized effect sizes mainly because when looking at the differential SCR data reported the authors observe M=0.31 and M=0.43 on day 1 and M=0.36 and M=0.17 on day 3 for active and sham respectively.

2.       Relevant for the interpretation of findings is that the authors may want to better explain the role of the stimulated brain region, DLPFC - MVPFC. Both in the introduction and discussion the authors mention that the right DLPFC was chosen because of its role in the processing of negative emotions. However, LeDoux and Pine (AJP in Advance; doi: 10.1176/appi.ajp.2016.16030353) very recently published a relevant paper on the distinction that emotions, feelings (anxiety), and defensive behaviors (bodily reactions to potential threat) are not the same thing with different underlying neural substrates. Given that this study targets a brain region thought to be associated with emotion processing and test the effects of stimulation on defensive behaviors, I suggest the authors include more theoretical discussion to address this issue. Moreover, the interpret of current, and previous, findings in light of hemispheric dominance for emotion processing theories in Discussion is somewhat unexpected. Mostly because these data may not in fact shed light on either of these theories as different types of emotion processing are tested. Hemispheric dominance experiments often make use of tasks that involve conscious processing of affective cues or visual manipulations to subliminal affective priming, whereas the current experiment tests defensive bodily reactions. I therefore would suggest limiting this possible explanation for the observed null findings.

3.       Instead, what I found missing in the Discussion, and to some extent in the introduction, is a better comparison of this study methods to prior work. The authors mention three prior tDCS studies on fear-related processes. Although relevant I think it crucial that the authors highlight some important differences in stimulation parameters in relation to the task. As we know tDCS can have different effects depending on whether tDCS was applied offline or online, stimulus intensity used, electrode size used and most importantly perhaps, electrode placement and when stimulation occurred in relation to fear processes. More specifically, in this paper as well as their previous paper the authors target reconsolidation of conditioned fear. On the other hand, Asthana et al. 2013 targeted consolidation of conditioned fear. Finally, van ‘t Wout et al. 2016 targeted extinction of conditioned fear. These are all very different manipulations and time points related to conditioned fear memory processes. Moreover, all three prior studies targeted different brain regions for stimulation, which is currently not clearly mentioned in the manuscript, but taken together with stimulation timing in may inform brain stimulation practices for ultimately clinical practice. I therefore think it is also important for the authors to elaborate on which other brain regions could have been implicated and what that may mean for the fear memory process they tested, reconsolidation.

Specific comments:

4.       Line 92: Why was a 38% reinforcement rate chosen?

5.       Line 96/97: 26 trials with 16 CS+ and 10 CS-, why a difference in the number of CS+ and CS-?

6.       tDCS methods: How much saline was used per sponge side? This may come across as a minute question, but recent discussion has been sparked regarding the saturation and preparation of tDCS sponges for stimulation. Also please add current density for ease of comparison to the literature.

7.       tDCS Methods: Did the authors collect data on impedance of tDCS?

8.       Line 104/105: Was the CS+ on day 2 combined with shock or not. From the sentence “On the second day all subjects were shown a reminder using a single presentation of the coloured square paired with the shock on Day 1 (CS+).” it is not clear whether the shock was delivered or not.

9.       Data Analysis: I fully understand the frustrating aspect of participants not conditioning properly. However, I wonder whether it is possible to use additional measures to define whether or not a participant conditioned adequately, as elimination 11 out of 25 (44%) participants seems somewhat high. Specifically, when looking at Milad et al., 2005 (in psychophysiology; DOI: 10.1111/j.1469-8986.2005.00302.x), they report that the difference in skin conductance response to the CS+ as compared to the CS- seem not to be greatest at the last trial of conditioning. In fact, it seems as if human participants, unlike rodents, habituate to the CS+ during conditioning and that fear responses reflected in SCR might be better observed in the earlier CS+ trials, at least in the paradigm used by Milad et al. 2005. So my question is would the group sizes change if the authors would use an averaged SCR over all CS+ trials as compared to all CS- trials during conditioning (minus the first trial)? Although I realize that this deviates from the methods these authors used in their prior tDCS study.

10.   Line 137: The authors use the differential SCR in the 0.5 to 4.5 second time window after stimulus onset. What was the baseline EDA assessment window to calculate the SCR? Was it the EDA right before the stimulus onset or EDA assessed during a baseline window prior to starting the experimental task?

11.   Results, line 146-149: Why did the authors perform a one-sample t-test of the last three trials and then perform an independent t-test of the last trial to test for active vs sham group differences. This seems to depart from what the authors write under Data Analysis where only “the last trial during fear acquisition on Day 1 was used as a criterion to decide if the subjects had successfully acquired fear conditioning”. Also I presume this last trials was a CS+ trial? Which then sparks the question whether the last three trials used for the one-sample t-test were the averages across the last three CS+ and the last three CS- trials, or instead, whether two out of the last three trials were a CS+ (or CS-) and one trial out of the last three trials was a CS- (or CS+)? In a nutshell, it isn’t entirely clear what the authors mean by the last (three) trials. The answer to this question may inform the analysis approach, and if the former is the case (the last three CS+ trials and the last three CS- trials), I wonder why the authors did not perform a repeated measures ANOVA to test for significant main effects of CS trial (CS+ vs CS-), group (tDCS vs sham), and the interaction between these variables. The same questions apply to the analyses for Day 3 data (Line 153).

12.   Results: Did the authors collect EDA during Day 2 to examine whether or not participants generated an SCR to the CS+?

13.   Results, lines 153-156: On Day 3 was there a significant difference between the CS+ and CS-? This would be important to report, preferably using a repeated measures ANOVA to again test for significant main (CS; group) and interaction effects (CS by group). This analysis might be (more) appropriate to test whether or not participants may still exhibit defensive responses or whether some habituation/extinction in fact took place between day 1 and day 3 and how this interacted with receiving sham or tDCS.

Author Response

Thank you for the valuable feedback and comments which we integrated in the revised manuscript. Please find below a point-by-point-reply.

The authors present a study in which they aim to inhibit defensive responses after fear reconsolidation in a Pavlovian fear conditioning paradigm using cathodal tDCS. This is in essence the opposite from an earlier study published by the authors where they used anodal tDCS to enhance reconsolidation after fear conditioning. Contrary to their previous finding of anodal tDCS, cathodal tDCS after fear reconsolidation did not seem to affect defensive responses 24 hours later. I think the publication of null results like this are important, because this would ultimately result in defining the parameters to allow better targeting of brain circuitry and processes for ultimate improved treatment of specific mental health problems. However, there are a number of issues I think the authors should address before I can recommend publication. In a way my comments are mostly related to 1) thinking about the data, analyses and subsequent interpretation, and 2) comparisons with prior studies – as that would allow a paper with non-significant results to have the largest impact on the field and accomplish the goal above.  

Broad comments:

1. Although the authors recognize and mention their small sample size; a total of 14 people – 7 randomized to active tDCS and 7 randomized to sham, this does remain a critical issue when dealing with interpreting results. I won’t ask for retrospective power analyses as I tend to find these circular in nature. However, I think it might be worthwhile to add standardized effect sizes mainly because when looking at the differential SCR data reported the authors observe M=0.31 and M=0.43 on day 1 and M=0.36 and M=0.17 on day 3 for active and sham respectively.

 As suggested by the reviewer, we have added standardized effect sizes (partial eta square) from the repeated measures ANOVA for all three factors (CS, group, CS x group interaction) for Day 1 and Day 3 to the paper (lines 160-173). Also, the sample size has increased slightly (n = 17) after implementing the reviewer’s suggestion of using the average of CS+ trials on Day 1 as a criterion for fear conditioning. Nevertheless, we concede that the sample size is small and mention it as a limitation of our study.

2. Relevant for the interpretation of findings is that the authors may want to better explain the role of the stimulated brain region, DLPFC - MVPFC. Both in the introduction and discussion the authors mention that the right DLPFC was chosen because of its role in the processing of negative emotions. However, LeDoux and Pine (AJP in Advance; doi: 10.1176/appi.ajp.2016.16030353) very recently published a relevant paper on the distinction that emotions, feelings (anxiety), and defensive behaviors (bodily reactions to potential threat) are not the same thing with different underlying neural substrates. Given that this study targets a brain region thought to be associated with emotion processing and test the effects of stimulation on defensive behaviors, I suggest the authors include more theoretical discussion to address this issue. Moreover, the interpret of current, and previous, findings in light of hemispheric dominance for emotion processing theories in Discussion is somewhat unexpected. Mostly because these data may not in fact shed light on either of these theories as different types of emotion processing are tested. Hemispheric dominance experiments often make use of tasks that involve conscious processing of affective cues or visual manipulations to subliminal affective priming, whereas the current experiment tests defensive bodily reactions. I therefore would suggest limiting this possible explanation for the observed null findings.

We have modified the introduction accordingly and expanded it by including a neuroanatomical discussion and cited recent papers by Otis et al. [1] and Hartley et al. [2] describing the neural circuit underlying fear memory reconsolidation and the connections between the prefrontal cortex and amygdala (lines 59-79). We have also expanded the discussion and cited the very recently published paper by LeDoux et al. [3] which could explain why targeting the prefrontal cortex with tDCS might only influence the cognitive circuit responsible for fearful feelings, but not necessarily the defensive survival circuit in the amygdala, which is responsible for physiological responses eg. skin conductance in case of our study (lines 211-222). We believe that discussing the two-circuit model proposed by LeDoux et al. in this context has important implications for developing protocols for future studies which manipulate fear using tDCS. We agree with the reviewer that the studies related to hemisphere dominance deal with tasks different from our experiment, we have limited this explanation in our Discussion section.

3. Instead, what I found missing in the Discussion, and to some extent in the introduction, is a better comparison of this study methods to prior work. The authors mention three prior tDCS studies on fear-related processes. Although relevant I think it crucial that the authors highlight some important differences in stimulation parameters in relation to the task. As we know tDCS can have different effects depending on whether tDCS was applied offline or online, stimulus intensity used, electrode size used and most importantly perhaps, electrode placement and when stimulation occurred in relation to fear processes. More specifically, in this paper as well as their previous paper the authors target reconsolidation of conditioned fear. On the other hand, Asthana et al. 2013 targeted consolidation of conditioned fear. Finally, van ‘t Wout et al. 2016 targeted extinction of conditioned fear. These are all very different manipulations and time points related to conditioned fear memory processes. Moreover, all three prior studies targeted different brain regions for stimulation, which is currently not clearly mentioned in the manuscript, but taken together with stimulation timing in may inform brain stimulation practices for ultimately clinical practice. I therefore think it is also important for the authors to elaborate on which other brain regions could have been implicated and what that may mean for the fear memory process they tested, reconsolidation.

We have added a new paragraph to the Discussion and compared the three different studies influencing fear with tDCS till date by Mungee et al.[4], Asthana et al.[5] and van’t Wout et al. [6] in detail including current density, stimulation location, process targeted (extinction/reconsolidation/consolidation), kind of stimuli used and how they affected fear (lines 223-245). We believe that the readers now get a sound overview of the stimulation protocols used and the effects that they have had, which in turn will help in developing brain stimulation practices for studies targeting fear and mental health disorders in future.  In addition, we discussed in more detail the neuronal networks involved in the assumed mechanism of action in our tDCS approach (lines 60-72).

Specific comments:

4.       Line 92: Why was a 38% reinforcement rate chosen?

We chose a partial reinforcement schedule during fear conditioning to try and avoid rapid extinction, which is probably more likely to occur if a 100% reinforcement schedule is used [7]. Partial reinforcement has the advantage of making fear acquisition slower and making the task for the test subjects non-trivial with an element of surprise, since not all CS trials are associated with the shock [8].

5. Line 96/97: 26 trials with 16 CS+ and 10 CS-, why a difference in the number of CS+ and CS-?

The 6 extra CS+ presentations were paired with the US (shock) in order to achieve 38% reinforcement rate. If we leave these 6 US trials out then the number of CS+ and CS- was equal (10). We adapted this protocol from the reconsolidation experiment conducted by Schiller et al. [9]

6.       tDCS methods: How much saline was used per sponge side? This may come across as a minute question, but recent discussion has been sparked regarding the saturation and preparation of tDCS sponges for stimulation. Also please add current density for ease of comparison to the literature

.

Unfortunately, we did not collect data on how much saline was used per sponge side. We concede the reviewer’s concerns that oversaturation of the sponge can result in increased area of current delivery leading to unintended variation between subjects. [10]. For future studies, we will follow a protocol with quantification of saline through syringes. We would like to thank the reviewer for making us aware of this important methodological point. We have added the current density (0.67 A/m2) to the methods section of the manuscript (line 120).

7.       tDCS Methods: Did the authors collect data on impedance of tDCS?

The maximum impedance during the experiments was 60 kΩ, however we did not collect individual data with respect to impedance.

8.       Line 104/105: Was the CS+ on day 2 combined with shock or not. From the sentence “On the second day all subjects were shown a reminder using a single presentation of the coloured square paired with the shock on Day 1 (CS+).” it is not clear whether the shock was delivered or not.

No shock was delivered on Day 2. We have reframed this sentence to make it clear to the readers (line 118).

9.       Data Analysis: I fully understand the frustrating aspect of participants not conditioning properly. However, I wonder whether it is possible to use additional measures to define whether or not a participant conditioned adequately, as elimination 11 out of 25 (44%) participants seems somewhat high. Specifically, when looking at Milad et al., 2005 (in psychophysiology; DOI: 10.1111/j.1469-8986.2005.00302.x), they report that the difference in skin conductance response to the CS+ as compared to the CS- seem not to be greatest at the last trial of conditioning. In fact, it seems as if human participants, unlike rodents, habituate to the CS+ during conditioning and that fear responses reflected in SCR might be better observed in the earlier CS+ trials, at least in the paradigm used by Milad et al. 2005. So my question is would the group sizes change if the authors would use an averaged SCR over all CS+ trials as compared to all CS- trials during conditioning (minus the first trial)? Although I realize that this deviates from the methods these authors used in their prior tDCS study.

We are very thankful to the reviewer for this suggestion and the interesting paper authored by Milad et al. [11]. Using the averaged scores for CS+ and CS- the sample size increased from 14 to 17 (real = 7, sham=10).

10.   Line 137: The authors use the differential SCR in the 0.5 to 4.5 second time window after stimulus onset. What was the baseline EDA assessment window to calculate the SCR? Was it the EDA right before the stimulus onset or EDA assessed during a baseline window prior to starting the experimental task?

We used Ledalab (a MATLAB based software), more specifically the CDA (Continuous Decomposition Analysis) method to analyze the skin conductance data. This method extracts the phasic information underlying the skin conductance response, and aims at retrieving the signal characteristics of the underlying sudomotor nerve activity [12]. In the case of CDA, the original EDA signal during the whole duration of the experiment is split into two components: tonic baseline and phasic SCR. The baseline EDA assessment window is the whole duration of the experiment in this case.

11.   Results, line 146-149: Why did the authors perform a one-sample t-test of the last three trials and then perform an independent t-test of the last trial to test for active vs sham group differences. This seems to depart from what the authors write under Data Analysis where only “the last trial during fear acquisition on Day 1 was used as a criterion to decide if the subjects had successfully acquired fear conditioning”. Also I presume this last trials was a CS+ trial? Which then sparks the question whether the last three trials used for the one-sample t-test were the averages across the last three CS+ and the last three CS- trials, or instead, whether two out of the last three trials were a CS+ (or CS-) and one trial out of the last three trials was a CS- (or CS+)? In a nutshell, it isn’t entirely clear what the authors mean by the last (three) trials. The answer to this question may inform the analysis approach, and if the former is the case (the last three CS+ trials and the last three CS- trials), I wonder why the authors did not perform a repeated measures ANOVA to test for significant main effects of CS trial (CS+ vs CS-), group (tDCS vs sham), and the interaction between these variables. The same questions apply to the analyses for Day 3 data (Line 153).

As suggested by the reviewer, we have performed a repeated measures ANOVA for the last three CS+ and the last three CS- trials. We found significant main effects of CS trial [F (1,15) = 15.22, p = 0.001] but no significant effects for group [F (1,15) = 1.09, p > 0.05] and the interaction between group and CS [F (1,15) = 4.19, p > 0.05]. The significant difference between CS+ and CS- indicates that participants successfully acquired fear conditioning on Day 1, and there was no significant difference between the sham and the real group (lines 157-164).

12.   Results: Did the authors collect EDA during Day 2 to examine whether or not participants generated an SCR to the CS+?

Unfortunately, we did not collect EDA on Day 2. In retrospect, it would have been helpful to check on Day 2 if the participants showed a response to the reminder. We acknowledge the reviewer’s suggestion for future studies.

13.   Results, lines 153-156: On Day 3 was there a significant difference between the CS+ and CS-? This would be important to report, preferably using a repeated measures ANOVA to again test for significant main (CS; group) and interaction effects (CS by group). This analysis might be (more) appropriate to test whether or not participants may still exhibit defensive responses or whether some habituation/extinction in fact took place between day 1 and day 3 and how this interacted with receiving sham or tDCS.

We acknowledge the reviewer’s suggestion and report results from the repeated measures ANOVA in the results section (lines 165-177). On Day 3, for the first three trials of CS+ and CS-, there were no significant effects for CS [F (1,15) = 2.05, p > 0.05] or group (tDCS/sham) [F (1,15) = 3.38, p > 0.05], the interaction between CS and group was also not significant [F (1,15) = 0.55, p > 0.05]. Additionally, we conducted a repeated measures ANOVA for only the first two trials of CS+ and CS- on Day 3. We found significant effects for CS [F (1,15) = 5.28, p < 0.05, ηp² = 0.26], but no significant effects for group [F (1,15) = 3.60, p > 0.05, ηp² = 0.19] or the interaction between CS and group [F (1,15) = 2.15, p > 0.05, ηp² = 0.13]. These results indicate that the participants showed defensive responses up to the first two trials of CS+ and CS-, tDCS had no effect on these defensive responses and rapid habituation took place between the second and the third trials so that there were no significant defensive responses after three trials each of CS+ and CS-.

References:

[1]          J.M. Otis, C.T. Werner, D. Mueller, Noradrenergic regulation of fear and drug-associated memory reconsolidation, Neuropsychopharmacology. 40 (2015) 793–803. doi:10.1038/npp.2014.243.

[2]          C.A. Hartley, E.A. Phelps, Changing Fear: The Neurocircuitry of Emotion Regulation, Neuropsychopharmacology. 35 (2010) 136–146. http://www.pubmedcentral.nih.gov/articlerender.fcgi?artid=3055445&tool=pmcentrez&rendertype=abstract.

[3]          J.E. Ledoux, Using Neuroscience to Help Understand Fear and Anxiety: A Two-System Framework, Am. J. Psychiatry. (2016). doi:10.1176/appi.ajp.2016.16030353.

[4]          A. Mungee, P. Kazzer, M. Feeser, M. a Nitsche, D. Schiller, M. Bajbouj, Transcranial direct current stimulation of the prefrontal cortex: a means to modulate fear memories., Neuroreport. (2013) 1–5. doi:10.1097/WNR.0000000000000119.

[5]          M. Asthana, K. Nueckel, A. Mühlberger, D. Neueder, T. Polak, K. Domschke, J. Deckert, M.J. Herrmann, Effects of transcranial direct current stimulation on consolidation of fear memory, Front. Psychiatry. 4 (2013) 1–7. doi:10.3389/fpsyt.2013.00107.

[6]          M. van ’t Wout, T.Y. Mariano, S.L. Garnaat, M.K. Reddy, S.A. Rasmussen, B.D. Greenberg, Can Transcranial Direct Current Stimulation Augment Extinction of Conditioned Fear?, Brain Stimul. (2016). doi:10.1016/j.brs.2016.03.004.

[7]          K.S. LaBar, J.C. Gatenby, J.C. Gore, J.E. LeDoux, E. a Phelps, Human amygdala activation during conditioned fear acquisition and extinction: a mixed-trial fMRI study., Neuron. 20 (1998) 937–945. doi:10.1016/S0896-6273(00)80475-4.

[8]          S. Meir Drexler, C.J. Merz, T.C. Hamacher-Dang, O.T. Wolf, Cortisol effects on fear memory reconsolidation in women., Psychopharmacology (Berl). (2016). doi:10.1007/s00213-016-4314-x.

[9]          D. Schiller, M.-H. Monfils, C.M. Raio, D.C. Johnson, J.E. Ledoux, E.A. Phelps, Preventing the return of fear in humans using reconsolidation update mechanisms., Nature. 463 (2010) 49–53. doi:10.1038/nature08637.

[10]        A.J. Woods, A. Antal, M. Bikson, P.S. Boggio, A.R. Brunoni, P. Celnik, L.G. Cohen, F. Fregni, C.S. Herrmann, E.S. Kappenman, H. Knotkova, D. Liebetanz, C. Miniussi, P.C. Miranda, W. Paulus, A. Priori, D. Reato, C. Stagg, N. Wenderoth, M.A. Nitsche, A technical guide to tDCS, and related non-invasive brain stimulation tools, Clin. Neurophysiol. 127 (2016) 1031–1048. doi:10.1016/j.clinph.2015.11.012.

[11]        M.R. Milad, S.P. Orr, R.K. Pitman, S.L. Rauch, Context modulation of memory for fear extinction in humans, Psychophysiology. 42 (2005) 456–464. doi:10.1111/j.1469-8986.2005.00302.x.

[12]        M. Benedek, C. Kaernbach, A continuous measure of phasic electrodermal activity, J. Neurosci. Methods. 190 (2010) 80–91.

Round 2

Reviewer 2 Report

The authors made some significant changes in their statistical approach and clarified and/or elaborated on other important aspects related to the interpretation of the presented data in the introduction and discussion. I would recommend acceptance of this manuscript.